

# Density and population viability of coastal marten: a rare and geographically isolated small carnivore

Mark A. Linnell[1,*], Katie Moriarty[2,*], David S. Green[3] and Taal Levi[4]

[1] Department of Forest Engineering, Resources, and Management, Oregon State University, Corvallis, OR, United States of America
[2] Pacific Northwest Research Station, United States of America Department of Agriculture, Forest Service, Olympia, WA, United States of America
[3] Institute for Natural Resources, Oregon State University, Corvallis, OR, United States of America
[4] Department of Fisheries and Wildlife, Oregon State University, Corvallis, OR, United States of America
[*] These authors contributed equally to this work.

Corresponding author
Katie Moriarty,
ktmoriarty22@gmail.com

## ABSTRACT

Pacific martens (*Martes caurina humboldtensis*) in coastal forests of Oregon and northern California in the United States are rare and geographically isolated, prompting a petition for listing under the Endangered Species Act. If listed, regulations have the potential to influence land-use decisions on public and private lands, but no estimates of population size, density, or viability of remnant marten populations are available for evaluating their conservation status. We used GPS and VHF telemetry and spatial mark-resight to estimate home ranges, density, and population size of Pacific martens in the Oregon Dunes National Recreation Area, central coast Oregon, USA. We then estimated population viability at differing levels of human-caused mortality (e.g., vehicle mortality). Marten home ranges were small on average (females $= 0.8$ km$^2$, males $1.5$ km$^2$) and density ($1.13$ martens/$1$ km$^2$) was the highest reported for North American populations (*M. caurina*, *M. americana*). We estimated 71 adult martens (95% CRI [41–87]) across two subpopulations separated by a large barrier (Umpqua River). Using population viability analysis, extinction risk for a subpopulation of 30 martens, approximately the size of the subpopulation south of the Umpqua River, ranged from 32% to 99% with two or three annual human-caused mortalities within 30 years. Absent population expansion, limiting human-caused mortalities will likely have the greatest conservation impact.

## INTRODUCTION

Conserving wildlife while maintaining economic growth is one of the most pervasive conservation and policy challenges globally. This balance in the United States is enforced in part by the Endangered Species Act (ESA), which can regulate land-use on both public and private lands for the conservation of imperiled species. Forests of the Pacific Northwest of North America highlight challenges between land-use and endangered species
conservation as demonstrated by the history with northern spotted owl (*Strix occidentalis caurina*, *Simberloff, 1987*). Now, decades after the conflict over listing the northern spotted owl, a distinct population segment of a forest-dependent small carnivore is a litigation target, petitioned for listing under the ESA (*Anonymous, 2017*).

Pacific martens (*Martes caurina*) are a small carnivore considered to be a habitat specialist closely associated with structurally complex montane forests with seasonal snow cover in the western United States (*Buskirk & Ruggiero, 1994*; *Zielinski, 2013*). Coastal populations of Pacific martens in Oregon and California, referred to as Humboldt marten (*Martes caurina humboldtensis*), are near the southern edge of their distribution and live in near-coast forests with limited or no snow cover. Recent extensive distributional surveys suggest two or three potential populations in coastal Oregon and northern California (*Moriarty et al., 2016*; *Zielinski et al., 2001*). These coastal populations of martens have contracted in the 20th century (*Zielinski et al., 2001*), prompting petitions to list a Distinct Population Segment as threatened or endangered (*Center for Biological Diversity, 2010*). The northernmost population is located in the central Oregon coast (*Moriarty et al., 2016*), and it is also the most isolated (i.e., >60 km from the nearest adjacent population).

The United States Fish and Wildlife Service determined that the coastal Distinct Population Segment of the Pacific martens in California and Oregon did not warrant listing as a threatened or endangered species under the Endangered Species Act in 2015 (*US Fish Wildlife Service, 2015*). The finding by the US Fish and Wildlife service, however, included two assumptions for which updated information now exists: (1) that coastal martens were abundant in central Oregon from the relatively high number of road-killed individuals there in the past three decades, and (2) extensive Late-Seral Reserves on federal lands provided habitat for these martens (*Slauson, 2015*). Recent distributional surveys indicated this population likely occupies a <500 m wide band of young (i.e., <70 years old) forests growing on sand dunes along the margin of the Pacific Ocean west of Highway 101, and that there is no evidence of martens >3 km inland (Fig. 1, *Moriarty et al., 2016*). With so little known about martens in the central Oregon coast, research needs including basic attributes of the population, such as population size, are urgently needed to inform conservation decisions.

Our objectives were to describe marten density, population size, and population viability in the central Oregon coast. Specifically, we used spatial mark-resight (SMR) models to evaluate density in a portion of our study area and then we applied our density estimate to coastal forests west of Highway 101 where martens resided to estimate total population size. We assumed that, (1) forest characteristics were similar across our study area, and (2) because martens are highly territorial, density would be static across study areas if home range sizes were similar between individuals. We then used a population viability analysis to quantify the potential effects of human-caused mortality on marten numbers (e.g., legal trapping, vehicle strikes). Finally, because density and home range size are often correlated with foraging resources (*Kittle et al., 2015*; *Mattisson et al., 2016*), we compared density and home range sizes of martens in coastal Oregon to other North American populations (*Martes caurina*, *Martes americana*) to infer year-round food resource availability compared to other populations.

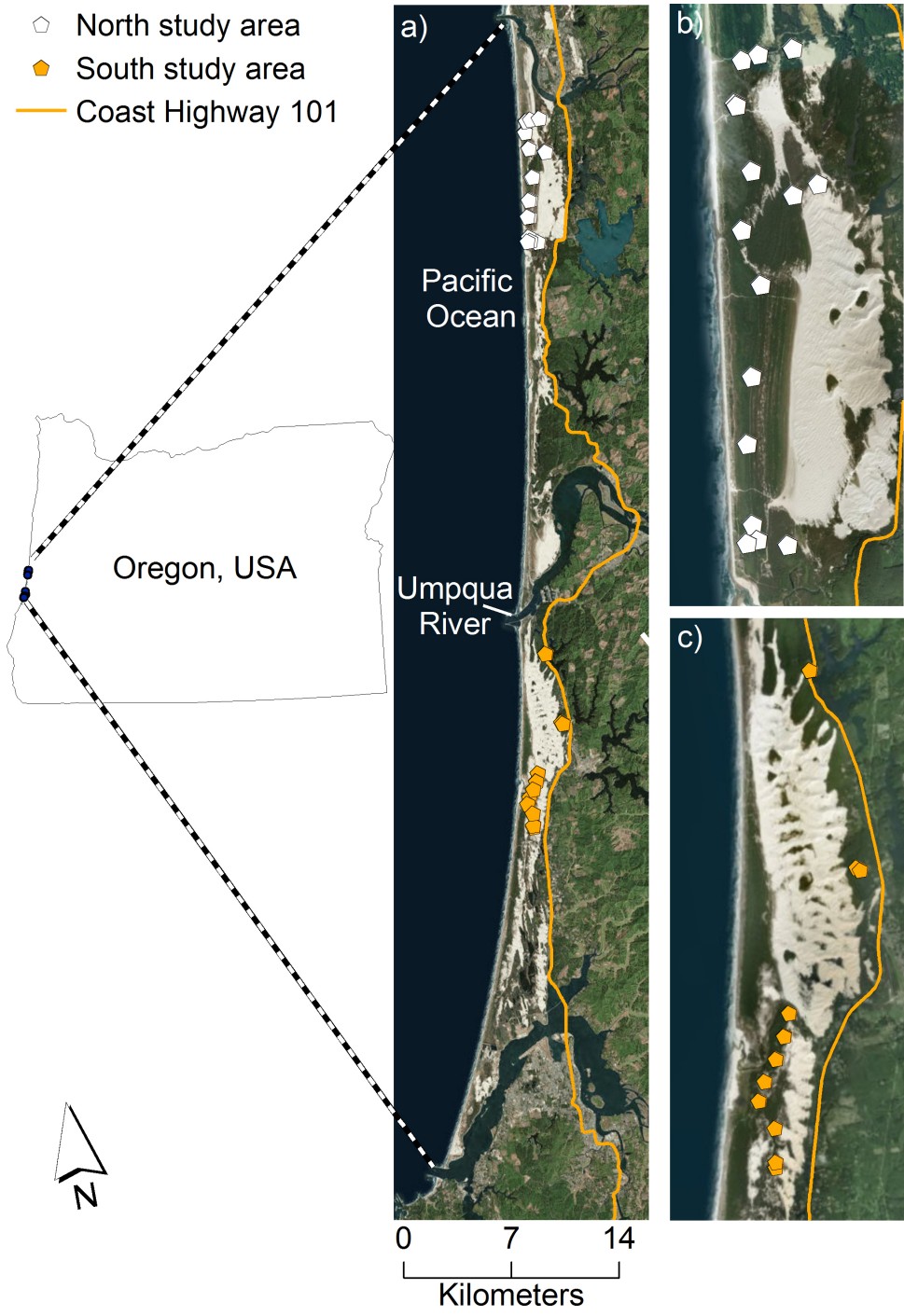

**Figure 1** **Our study area of coastal Pacific martens in the Oregon Dunes Recreation Area.** We collected location data on coastal Pacific martens (*Martes caurina humboldtensis*) in the Oregon Dunes Recreation Area, west of Highway 101, Oct. 2015 to Jan. 2016. The study area is bounded to the north and south by the Siuslaw and Coos Rivers, respectively, and divided by the Umpqua River in the center, which is approximately 600 m wide where it meets the Pacific Ocean. Live-trap locations are represented by the white and orange pentagons in the north and south study areas, respectively. The complete study area is displayed in (A), and insets of the north and south study areas, specifically, are displayed in (B) and (C). Image data: Esri, DigitalGlobe, GeoEye, Earthstar Geographics, CNES/Airbus DS, USDA, USGS, AeroGRID, IGN, and the GIS User Community.

## METHODS

### Study area

We surveyed the northernmost population of coastal martens along the central Oregon coast within the 125 km$^2$ Oregon Dunes National Recreation Area (hereafter, "Oregon Dunes"). Coastal forests within the Oregon Dunes consisted of a narrow north-south strip along the margin of the Pacific Ocean bounded by two large rivers to the north and south (i.e., Siuslaw and Coos Rivers), Highway 101 to the east, and bisected by the Umpqua River, which is 600-m wide at the confluence with the Pacific Ocean (Fig. 1). Much of the forested area was the result of recent expansion over the last 70 years coincident with stabilization of near-coast beaches by European beach grass (*Amophila arenaria*) into mounded fore dunes, which limited sand deposition and facilitated vegetation expansion into previously shifting open sand (*Christy, Kagan & Wiedemann, 1998*).

Coastal dune forests grew on nutrient poor sandy soils (*Christy, Kagan & Wiedemann, 1998*), and they were dominated by young (<70 years-old) shore-pine (*Pinus contorta contorta)* and Sitka spruce trees (*Picea sitchensis)*. The sub-tree canopy was dense, extended to >2.5 m in height, and it was dominated by willow (*Salix hookeri*), Pacific waxmyrtle (*Myrica californica*), salal (*Gaultheria shallon*), and slough sedge (*Carex obnupta*) on seasonally flooded sites, and berry-producing ericaceous shrubs (e.g., evergreen huckleberry *Vaccinium ovatum*, salal) on seasonally dry sites (*Christy, Kagan & Wiedemann, 1998*). Coastal forests differed substantially from inland forests east of Highway 101 in vegetation age, structure, composition, and their vertebrate communities (*Eriksson, 2016*). Inland forests were a mix of young (i.e., 0–80 years) and mature (i.e., >80 years old) Douglas-fir (*Pseudotsuga menziesii*) and Sitka spruce forests. Mature forests on federal lands were primarily managed as Late Successional Reserves to protect habitat for northern spotted owls (*Strix occidentalis)* and marbled murrelets (*Brachyramphus marmoratus*, *Davis et al., 2015*). Forests in the Oregon Dunes supported a high diversity of vertebrates, including several predators and competitors of martens (e.g., gray foxes *Urocyon cinereoargenteus*, coyotes *Canis latrans*, cougars *Puma concolor*; *Eriksson, 2016*).

To distinguish vegetation cover from open sand, we used airborne light detection and ranging data collected at 1-m resolution. We defined vegetation cover as >40% cover of pixels >1 m in height within a 100-m circular radius moving window of each pixel. This process produced a raster layer that smoothed small gaps in vegetation cover (i.e., sand gaps <30 m) that martens could presumably move through, but that excluded broad expanses of open sand that we assumed represented non-habitat for martens, particularly because of the presence of predators (*Moriarty et al., 2015*). The northern (i.e., north of Umpqua River, Fig. 1B) and southern (i.e., south of Umpqua River, Fig. 1C) study areas were comprised of 36.9 km$^2$ and 25.6 km$^2$ of vegetation cover, respectively.

Minimum and maximum temperatures in July and January were 10.1 °C and 20.3 °C and 3.2 °C and 10.2 °C, respectively. Annual precipitation averaged 176 cm, and occurred primarily between November and March (Western Regional Climate Center 1971–2016). Elevation within the study area ranged from eight to 80 m.

## Live-capture and home range size estimates

We live-trapped and radio-marked martens from October to December 2015 using traps spaced approximately 1 km apart with some additional clustering of traps at <1 km (Fig. 1) using methods described in *Moriarty et al. (2017)* and *Mortenson & Moriarty (2015)*. We fit adult martens (i.e., >two years old) with a VHF (Advanced Telemetry Systems, Minnesota, USA; 29 g) or GPS/VHF collar (Quantum 4000 Micro-Mini GPS collars, Telemetry Solutions, CA, USA; 41–44 g; or G10 snap technology GPS, Advanced Telemetry Systems, Isanti, MN, USA; 27 g; Table S1). Each individual marten was marked with a unique pattern of reflective tape attached to the antenna of the radio collar, which we used to resight marked individuals using black-LED remote cameras (Bushnell Aggressor, model: 119776; Bushnell Corporation, Overland Park, KS, USA; Fig. 2). Most martens (80%) were captured only once prior to the spatial mark-resight survey and then re-captured after this survey was complete when we removed collars in January and February 2016. All capture and handling procedures were approved by the USDA Forest Service's Institute for Animal Care and Use Committee (USFS 2015–002) under an Oregon Department of Fish and Wildlife Scientific Take Permit (ODFW 119–15).

We programmed GPS collars to collect locations separated by 5 min, and only included locations in our analyses with predicted errors <30 m and time periods where data were collected for >72 consecutive hrs (details in *Moriarty et al., 2017*). We located individuals with VHF-only collars at least twice per week. We only used VHF locations where the variance of x and y was <400 m determined with Location of a Signal (Ecological Software Solutions LLC). We estimated 99% local convex hull home ranges, discarding 1% of the furthest dispersed locations (*Lyons, Turner & Getz, 2013*), using the t-LoCoH package in R. Local convex hulls were constructed using 35 neighboring locations ($k = 35$), regardless of time between locations ($s = 0$; *Lyons, Turner & Getz, 2013*; *R Core Team, 2017*). These parameterizations best reflected marten space use in our study (*Moriarty et al., 2017*) by limiting the formation of multiple activity centers, and they provided a smoothed outer contour boundary.

## Spatial mark-resight

We monitored 31 sites for 39 consecutive nights in December 2015 and early January 2016 along a linear transect (henceforth, "SMR transect") that overlapped the area occupied by radio-tracked martens in the northern study area. Each site was distributed (mean ± 1 standard deviation) 311 ± 91 m apart within vegetation cover with a minimum goal of four sites accessible to each female (*Sun, Fuller & Royle, 2014*). Each site consisted of one remote camera placed 0.5 m high, 2–4 m from attractants (i.e., bait, olfactory lure) and a measuring strip, such that the camera field of view was centered on the attractants (Fig. 2). We placed bait, in the form of ~250 g of chicken and ~100 g of strawberry jam, at each site during setup and replaced it at each site on 3 visits. Visits occurred every (mean ± 1 standard deviation) 8.2 ± 1.5 days after setup. We reviewed photos for each site and occasion, which we defined as the 24-hour day, to identify marked or unmarked individuals. We had (mean ± 1 standard deviation [range]) 32.8 ± 67.0 [1–637] photos available per site and occasion to identify marked or unmarked individuals. We censored

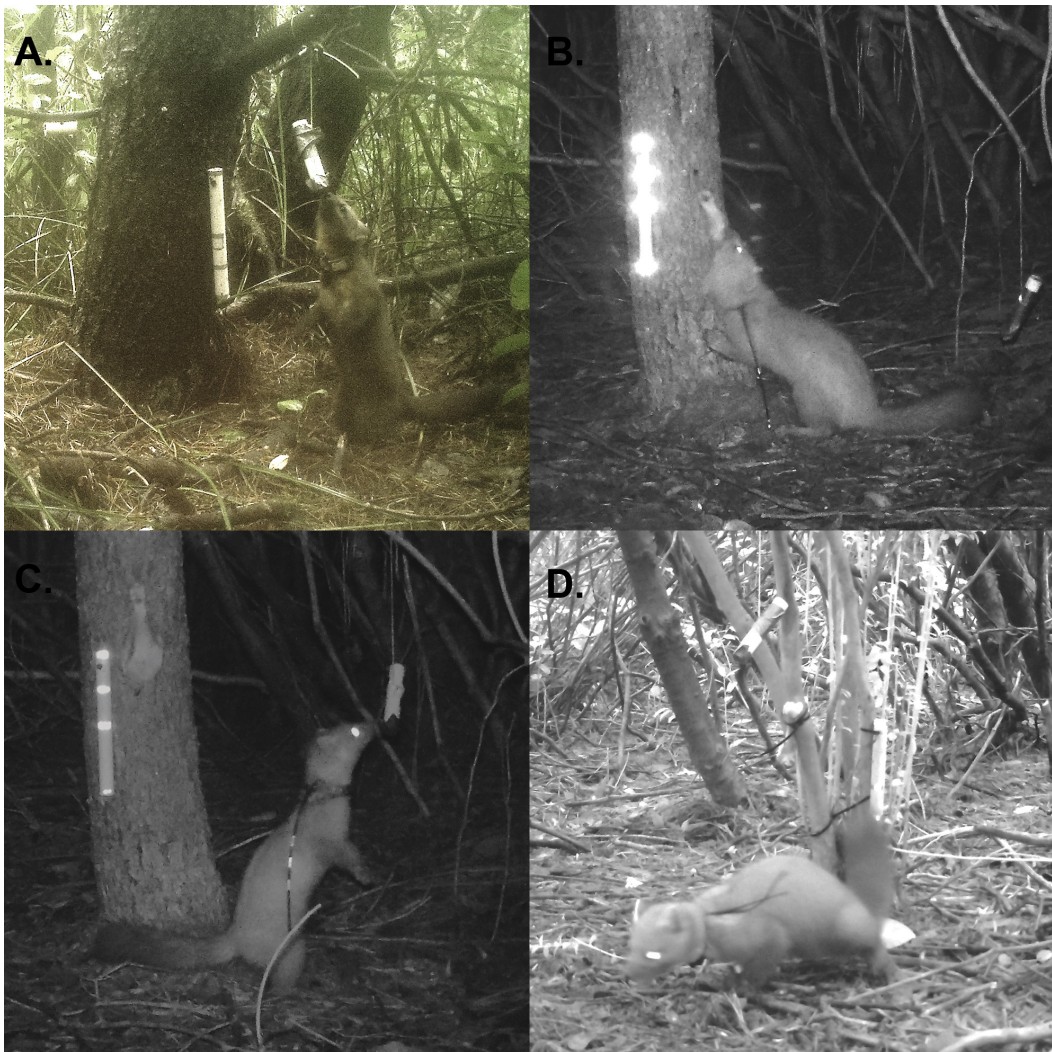

**Figure 2 Photographs of uniquely marked martens.** Examples of uniquely marked individual martens (*Martes caurina humboldtensis*). (A) A female marten with GPS collar sniffing strawberry jam. (B) A male marten with two reflective bands (middle, end of antenna). (C) A female marten with three reflective bands. (D) A male marten with unique GPS collar with two antennas. Each station included an olfactory lure (Gusto, Minnesota Trapline Products, Pennacock, MN) and baits that were checked and replaced weekly. We set remote cameras (Bushnell Aggressor, model: 119776; Bushnell Corporation, Overland Park, KS, USA) 2–4 m from bait and we programmed them to record one photo after motion was detected with a one-second lag between consecutive photos.

photos that we were unable to determine if the marten was marked or unmarked (<1% of all photos).

We estimated the density of martens from our photographic data using a generalized spatial mark-resight model (*Whittington, Hebblewhite & Chandler, 2018*; Code in Article S1) based on the methods of spatial capture-recapture (*Efford, 2004*; *Royle & Young, 2008*). Generalized spatial mark-resight models combine the latent processes that generate both the capture (i.e., marking) and resight data (i.e., cameras) to estimate the number of latent

activity centers ($s_i$) within the study area (*Whittington, Hebblewhite & Chandler, 2018*). We defined our study area as a discrete state-space $S$ of a 100-m grid within a 5-km buffer around camera stations, excluding cells in the Pacific Ocean.

We defined live-capture data as the binomially distributed random variable $ycap_{ij}$ representing the number of times that marten $i$ was captured in trap $j$ as a function of the probability of capture ($pcap_{ij}$) and the number of nights that trap $j$ was open ($Ktrap_j$):

$$ycap_{ij} \sim Binom(Ktrap_j, pcap_{ij}).$$

We hypothesized that the probability of capture would vary by sex and the distance between the location of trap $j$ and the activity center of marten $i$:

$$pcap_{ij} = p0cap_i \times e^{(-d_{ij}^2/2\sigma_k^2)}$$

where the average probability of capture $p0cap_i$ was modeled as a function of the sex of each marten ($logit(p0cap_i) = \beta_0 + \beta_1 \times sex_i$), a half-normal decay function where $d_{ij}$ is the distance between the trap and the latent activity center of individual $i$ ($s_i$), and the standard deviation of a bivariate normal distribution reflecting space-use varying by sex ($\sigma_k$). We parameterized $\sigma$ separately for each sex (See Article S1).

We defined camera resighting data as the Bernoulli distributed random variable $ycam_{ijk}$ representing whether or not the previously live-captured marten $i$ was resighted at camera station $j$ on occasion $k$ as:

$$ycam_{ijk} \sim Bern(pcam_{ijk})$$

where $pcam_{ijk}$ is a function describing the average daily rate of detecting martens on camera. Similar to the capture data, we hypothesized that the average daily rate of detection would vary by sex and the distance between the station and their latent activity center ($pcam_{ijk} = p0cam_{ijk} \times e^{(-d_{ij}^2/2\sigma_k^2)}$). We also hypothesized that the time since bait addition would influence the rate of detecting martens on camera, so we added a variable to test the effect of days since baiting at station $j$ on occasion $k$ ($days_{jk}$) = ($logit(p0cam_{ijk}) = \delta_0 + \delta_1 \times sex_i + \delta_2 \times days_{jk}$).

We modeled activity center locations using a non-homogeneous Poisson point process in $S$ to examine whether locations of marten activity centers in the Oregon Dunes were associated with percent vegetation cover. We calculated the percent vegetation cover in each grid cell $g$ in $S$, and used an intensity function following *Royle et al. (2014)* to model the probability of an individual being in grid cell $g$ ($p_g$) as:

$$p^g = \mu_g/EN$$

where $\mu_g$ is a function of an intercept ($\alpha_0$), the linear effect of vegetation cover ($\alpha_1$), and the size of the grid cell ($area_g$; $\mu_g = area_g \times e^{a_0 + a_1 \times vegetation\ cover_g}$), and is divided by the expected number of martens in the study area ($EN$). We incorporated telemetry data to increase the precision of our estimates for the movement parameters and the location of marten activity centers (*Royle et al., 2014*; *Sollmann et al., 2013*). Telemetry locations of martens were modeled as being generated from a bivariate normal movement model with

a mean of $s_i$ and a precision of $\frac{1}{\sigma_k^2}$ (See Article S1). Martens can travel to any point in their home range within one hr (*Moriarty, Epps & Zielinski, 2016*; *Moriarty et al., 2017*). Accordingly, we only used locations that were at least one hr apart to ensure independence of telemetry locations (*Sollmann et al., 2013*).

We followed *Whittington, Hebblewhite & Chandler (2018)* in their treatment of the sightings of unmarked individuals; the detections of unmarked individuals at camera station $j$ on occasion $k$ ($nU_{jk}$) was modeled as:

$$nU_{jk} \sim Sum(yu_{1jk}, yu_{2jk}, yu_{3jk...}yu_{ijk})$$

$$yu_{ijk} \sim Bern(pcam_{ijk})$$

where the number of sightings of unmarked individuals was modeled to be generated from a latent Bernoulli process of resight probability ($yu_{ijk} \sim Bern(pcam_{ijk})$), based on the same probabilities of resighting as defined previously ($pcam_{ijk}$). Unmarked individuals were seen infrequently on our cameras ($n = 14$ sightings throughout the duration of the study), and identified as being present at the camera for a single bout typically <19 min. Thus, it was highly unlikely that more than 1 unmarked individual was captured on our cameras per day. The code for our SMR model can be found in Article S1.

We fit our models using data augmentation (*Royle & Dorazio, 2008*; *Royle & Young, 2008*) and the Markov-Chain Monte Carlo (MCMC) methods of JAGS (*Plummer, 2003*) with the jagsUI package (*Kellner, 2014*) in R v. 3.4.3 (*R Core Team, 2017*). We used uninformative prior distributions for all parameters (See Article S1). We calculated estimates from 3,000 MCMC samples, taken from three chains run for 10,000 iterations, thinned by five, following a burn-in of 5,000. We assessed model convergence by examining trace plots and $\hat{R}$ values for parameter estimates (*Gelman & Hill, 2007*; *Gelman et al., 2014*). All $\hat{R}$ values were <1.1, indicating chain convergence. We estimated the density of martens in our northern study area by determining the number of martens with estimated activity centers located in vegetation cover in the state-space, excluding open sand.

To evaluate our assumption that home range sizes were similar, we compared home range sizes in the northern and southern study areas using a general linear model with two parameters: sex and study area. We interpreted test statistics from this model and lacking any significant differences, we assumed density could be extrapolated to estimate population sizes (*Moriarty et al., 2017*).

## Population viability

We assessed the risk of extirpation for a marten subpopulation over the next 30 years in the context of threats from human activities (e.g., trapping, roadkill; *Gerber, Buenau & Vanblaricom, 2004*). We estimated the maximum intrinsic population growth rate using a modified Euler-Lotka equation proposed by *Skalski, Millspaugh & Ryding (2008)*

$$e^{ra} - e^{-M}(e^r)^{a-1} - ml_a = 0,$$

where $r$ is the maximum intrinsic growth rate, $a$ is the age at first birth, $m$ is the fecundity constant (number of female offspring/female/year), $e^{-M}$ is the probability of survival, and $l_a$ is the probability of survival to maturity (survivorship). We obtained a range of parameter

**Table 1  Input values for coastal marten viability analysis.** Input values for coastal Pacific marten (*Martes caurina humboldtensis*) viability analysis in the Oregon Dunes Recreation Area.

| Variable | Value | Justification |
|---|---|---|
| Age at first parturition | 2 | *Mead (1994)* |
| Average number of kits/year (*m*) | 1.5[a] | *Aune & Schladweiler (1997)*, *Flynn & Schumacher (2016)* |
| Survivorship to first parturition (*la*) | 0.35 | |
| Kit survival (age 0–1) | 0.49 | *Johnson et al. (2009)* |
| Yearling survival (age 1–2) | 0.7 | Average for North American martens, *McCann, Zollner & Gilbert (2010)* |
| Range of adult survival (age 2+) | 0.7–0.9 | *McCann, Zollner & Gilbert (2010)* |

**Notes.**

[a]We choose $m = 1.5$ assuming three offspring and a 50% sex ratio as reasonable as among the highest observed litter size that would be expected to be achieved at low population density. For instance, *Strickland & Douglas (1987)* reported that both pregnancy rates and numbers of corpora lutea in pregnant female martens in Ontario were stable, ranging from 91–100% and 3.19–3.53, respectively. *Aune & Schladweiler (1997)* reported pregnancy rates similar for two populations in Montana, ranging from 76–95% over five years, but a lower mean number of corpora (2.6) per adult female in the southwestern part of the state leading to an estimate of $m = 1.1$. *Thompson & Colgan (1987)* reported 2.74–3.46 corpora lutea in pregnant females. *Flynn & Schumacher (2016)* observed pregnancy rates of martens in Southeast Alaska averaging only 47% over seven years while litter size was 3.3, producing an estimate of $m = 0.78$.

**Table 2  Bracketing uncertainty with three maximum intrinsic growth rates (*r*).** Three estimates of maximum intrinsic growth rate (*r*) for coastal Pacific martens (*Martes caurina humboldtensis*) bracketing our uncertainty from most to least conservative life history assumptions of annual survival for population viability modeling.

| Annual survival ($e^{-M}$) | Female kits per year (*m*) | Age of first parturition (*a*) | Survivorship to age at first parturition ($l_a$) | Maximum intrinsic growth rate (*r*) |
|---|---|---|---|---|
| 0.7 | 1.5 | 2 | 0.35 | 0.143 |
| 0.8 | 1.5 | 2 | 0.35 | 0.205 |
| 0.9 | 1.5 | 2 | 0.35 | 0.268 |

estimates associated with the maximum reproductive output of two closely related species of North American martens (*Martes americana, Martes caurina*) in wild populations from the literature (Table 1). Estimates of *r* are sensitive to uncertainty in annual survival, $e^{-M}$; we estimated maximum intrinsic growth rate assuming average, high, and very high survival rates (0.7, 0.8, 0.9 respectively, *McCann, Zollner & Gilbert, 2010*) to obtain three values of $r = 0.143, 0.205, 0.268$ (Table 2). We used the intermediate value of $r = 0.205$ in our population projections, but we also implemented a stochastic element with $\sigma = 0.06$ such that the low and high estimates of *r* would bracket one standard deviation from the mean.

We simulated the dynamics of a population beginning at carrying capacity using initial values of the population size (*K*) equal to 20, 30, and 40 to illustrate how estimates of extirpation risk depend on our uncertainty about the current population size, assuming that immigration between the northern and southern study areas was infrequent due to a large barrier (i.e., Umpqua River; see population estimates in *Results*). The density-dependent population dynamics are given by the discrete theta-logistic model with an annual mortality component:

$$N_{t+1} = e^{r\left(1-\left(\frac{N_t}{K}\right)^{\theta}\right)+\varepsilon} - H_t,$$

where $\varepsilon \sim N(0, \sigma)$, with $\sigma = 0.06$ based on the variation in our best estimate of $r$ (Table 2), and mortalities resulting from trapping and road-kills as $H_t \sim Pois(\lambda)$. The rate parameter of the Poisson distribution, $\lambda$, defines both the mean and variance of the annual mortality through road-kills or trapping ($H_t$), which takes values of $\lambda = 1$, 2, or 3 martens in our models (36 martens harvested 1969–1995, 0–4/year; *Verts & Carraway, 1998*). We assumed a small density-independent harvest to illustrate how extirpation risk can be influenced by relatively low levels of human-caused mortality. We conservatively assumed a standard logistic population growth ($\theta = 1$), but we also assumed that density-dependent declines in per-capita growth occurred at higher population densities ($\theta = 2$), which is expected for long-lived mammals (*Boyce, 1992*). We simulated 1,000 population trajectories for each of three initial conditions ($K = 20$, 30, and 40), three stochastic human-caused mortality rates ($\lambda = 1$, 2, and 3), and two values of the strength of density dependence using theta ($\theta = 1$, 2). Finally, we report observed mortalities during our study period. Where appropriate, we report results as mean $\pm$ 1 standard deviation.

## RESULTS

We live-captured and radio-collared seven females (four VHF-only, three GPS/VHF) and four male (all GPS) martens. Our GPS collars collected 1,139 ($\bar{x}$, range: 173–2,960) locations over 15.7 ($\bar{x}$, range: 4–44) days on 8 individuals (4 males, 4 females), and we collected 35 ($\bar{x}$, range: 23–37) locations over 75 ($\bar{x}$, range: 42–90) days for three females with VHF-only collars (Table S1). Home range sizes in our study areas were similar ($t = 0.5$, $p = 0.68$) for males: 1.7, 2.2 (northern, $n = 2$) vs 1.0, 2.2 (southern, $n = 2$), and females 0.59–0.84 ($\bar{x} = 0.67$, northern, $n = 4$) vs 0.71, 0.79 (southern, $n = 2$). Martens were primarily located in areas of high vegetation cover; vegetation cover within a 100 m moving window of telemetry locations averaged 75% (25–75% quantile range = 60–96%, $n = 11$ martens; Table S1, Fig. S1). Home range sizes were smaller and density was higher in coastal Oregon compared to other North American populations (Fig. 3, Table S2). Home range sizes were negatively correlated with density (Fig. 3).

### Density and population size

We incorporated 79.3 $\pm$ 59.2 telemetry locations per individual into our SMR models. No marked individuals of the same sex were observed visiting the same camera station, and 1.1 ($\bar{x}$, range: 0.7–1.9) km and 4.7, 2.1 ($n = 2$) km was the furthest distance between camera station detections for females and males, respectively (Fig. 4).

We estimated marten density as (mean $\pm$ 1 standard deviation) 1.13 $\pm$ 0.15 individuals/km$^2$ (95% CRI [0.81–1.39]), or 9.75 $\pm$ 1.32 individuals within the SMR area. Assuming density was constant within vegetation in the 62.5 km$^2$ Oregon Dunes, we estimated a median population size of 42 (CRI = 30–51) north of the Umpqua River and 29 (CRI = 21–36) south of it. Sex did not have a significant effect on the probability of live-capture (Table 3), but female martens had a higher resight probability than males

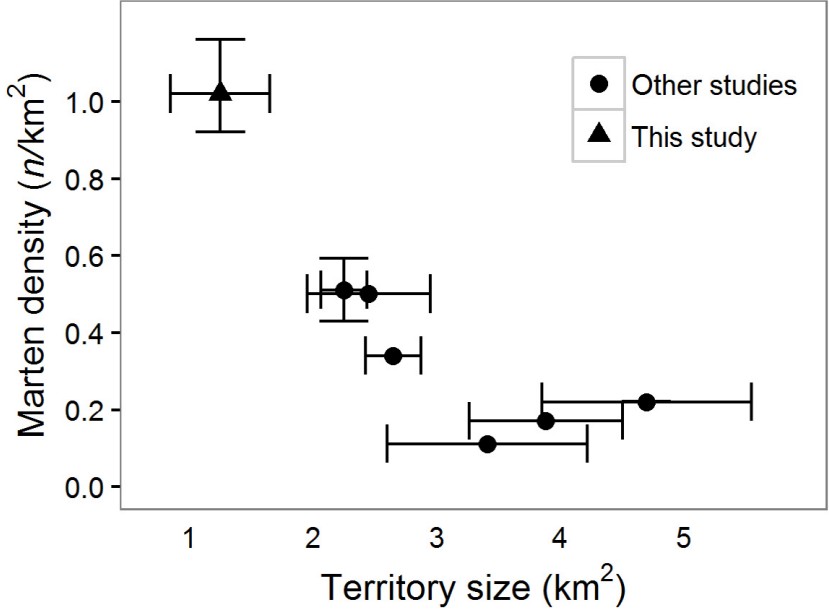

**Figure 3 Density and home range size of North American martens.** Technical articles that included the keywords "marten", "density", "territory", and "home range". Of the >75 papers reviewed for North American martens (*Martes americana, M. caurina*), four reported both home range sizes and density. Territories were estimated using either 100% Minimum Convex Polygons (MCP) or time-influenced Local Convex Hulls (t-LoCoH). Reported densities were either minimum known alive (MNKA) or calculated with spatial mark-resight. Mean and 95% confidence intervals reported if available in the study. Other studies were conducted in Maine, USA which included estimates from three study areas (*Payer & Harrison, 1999*), central British Columbia, Canada (*Poole et al., 2004*), New Hampshire, USA (*Sirén et al., 2016*), and Quebec, Canada (*Godbout & Ouellet, 2010*, Table S2).

(Table 3). Days since baiting had a significant effect on resight probability; martens were more likely to visit baited cameras closer to a baiting event (Table 3). Percent vegetation cover had a significant effect on the distribution of activity centers (Table 3).

## Population viability

We estimated that two or more annual human-caused mortalities on martens (e.g., trapping and road-kills) would lead to a substantial risk of extirpation, particularly at smaller population sizes (Figs. 5 and 6) and for $\theta = 1$ (Fig. 5) relative to $\theta = 2$ (Fig. 6). The likelihood of extirpation when $\theta = 1$ for a population of 30 individuals, which approximated the average of our estimates for each study area, was 32% and 99% with two and three annual mortalities, respectively. The probabilities decreased to 1% and 60% when $\theta = 2$ with two and three annual mortalities, respectively. The probability of extirpation increased to 89–100% and 65–100% for a population of 20 individuals with two or three annual mortalities.

## DISCUSSION

Our population assessment revealed that the central Oregon population of coastal martens contains fewer than 87 adults divided into two subpopulations separated by a riverine

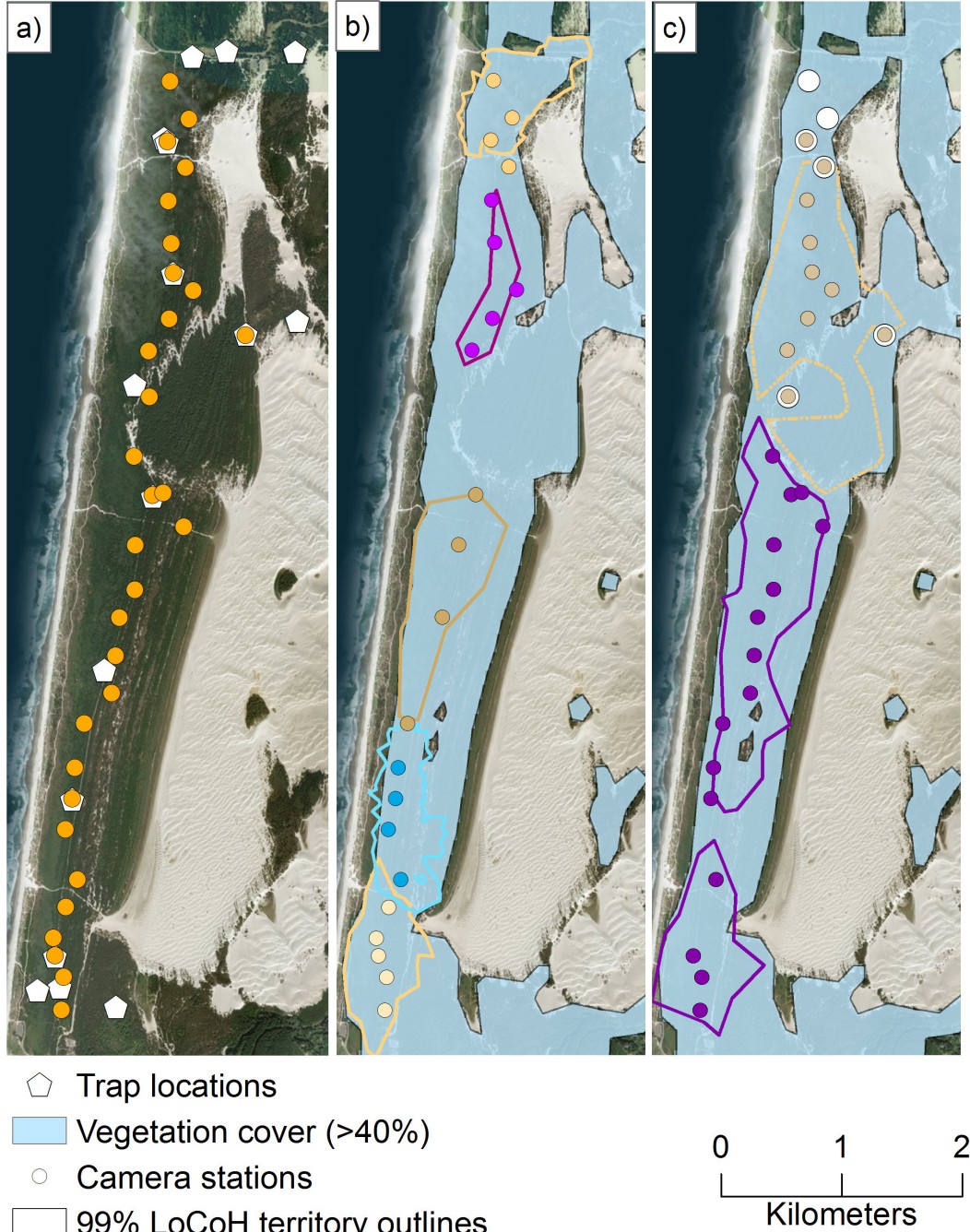

○⬠ Trap locations
▨ Vegetation cover (>40%)
○ Camera stations
▭ 99% LoCoH territory outlines

0    1    2
Kilometers

**Figure 4** **Our spatial mark-resight study area and coastal Pacific marten locations from remotely triggered cameras.** We conducted a spatial mark-resight study using remotely triggered cameras and by marking the coastal Pacific marten (*Martes caurina humboldtensis*) with unique reflective strips on their collars (Fig. 2) in the northern portion of the coastal Oregon Dunes National Recreation Area from 4 December 2015 to 12 January 2016. Here, we show the (A) locations of all traps for live-capture (pentagons) and camera stations (orange circles), (B) stations that detected female martens, (C) stations that detected male martens, and unmarked martens (large white circles). For (B) and (C), individual martens are depicted by unique colored dots (camera station detections), and outlines (outer boundary of territories). The light blue shading depicts vegetation >1 m in height and with >40% cover within a 100-m circular radius estimated from a light detection and ranging canopy height model. Image data: Esri, DigitalGlobe, GeoEye, Earthstar Geographics, CNES/Air- bus DS, USDA, USGS, AeroGRID, IGN, and the GIS User Community.

**Table 3 Summary statistics of marten population density and detection rates using a spatial mark-resight (SMR) model.** Summary statistics from a spatial mark-resight model with telemetry data that estimated the density of the Humboldt subspecies of Pacific martens (*Martes caurina humboldtensis*) in our study area in the Oregon Dunes Recreation Area from October 2015 to January 2016. Significant effects (parameters with 95% CRI's not-overlapping 0), not including estimates of density, abundance, sigma, or intercepts, are indicated in bold.

| Parameter | Mean (SD) | Credible Interval | | |
|---|---|---|---|---|
| | | 2.5 | 50 | 97.5 |
| Density (per km$^2$)[a] | 1.13 (0.15) | 0.81 | 1.15 | 1.39 |
| Abundance (# martens)[a] | 9.75 (1.32) | 7 | 10 | 12 |
| $\alpha_0$—habitat intercept | −1.55 (0.76) | −3.29 | −1.45 | −0.38 |
| $\alpha_1$—effect of forest cover | **1.07 (0.53)** | **0.17** | **1.02** | **2.22** |
| $\beta_0$—capture probability intercept | −1.91 (0.53) | −2.98 | −1.9 | −0.94 |
| $\beta_1$—female effect on capture probability | 0.79 (0.6) | −0.36 | 0.79 | 1.98 |
| $\delta_0$—resight probability intercept | −1.26 (0.17) | −1.59 | −1.26 | −0.92 |
| $\delta_1$—female effect on resight probability | **0.75 (0.19)** | **0.38** | **0.76** | **1.11** |
| $\delta_2$—days since baiting effect on resight probability | **−0.06 (0.02)** | **−0.11** | **−0.06** | **−0.02** |
| $\sigma_{male}$ (m) | 1,141.22 (45.27) | 1,058.39 | 1,139.46 | 1,233.75 |
| $\sigma_{female}$ (m) | 277.81 (6.17) | 266.46 | 277.63 | 290.1 |

Notes.
[a]Based on the habitat mask within our state-space.

barrier. Further, this population appears completely isolated with a lack of connectivity to the southern Oregon population. Based on the small number of individuals in these subpopulations, our projections suggest that even a small amount of human-caused mortalities will strongly increase the likelihood of extirpation over the next 30 years. Further, our analysis is likely an optimistic scenario for marten population viability because we assumed that marten populations would exhibit very high survival and fecundity at low population densities, which may not be the case. Despite these favorable assumptions, marten population viability was low given modest mortality estimates averaging 2–3 individuals annually, even when assuming higher than observed carrying capacities and assuming later onset of density dependence ($\theta = 2$; Fig. 6). Moreover, we did not fully consider environmental stochasticity or catastrophes in our viability analysis. In particular, the extant central Oregon coast marten population is in a tsunami zone within the Cascadia subduction zone. The probability of a large earthquake and tsunami eliminating much of Oregon's near-coastal forests in the next 50 years is placed at 15 to 20% (*Goldfinger et al., 2012*). Such an event would be expected to eliminate much of the forests that the central coast marten population occupies.

Martens can be common in structurally complex high elevation montane forests with seasonal snow cover, but they are apparently rare and geographically isolated in coastal Oregon. Nonetheless, the Oregon Dunes supported the smallest home ranges and highest reported density of martens in North America (Fig. 3). North American martens inhabiting forests with seasonal snow-cover typically consume a narrow range of prey, especially during winter months (*Martin, 1994*), which can lead to substantial inter- and intra-annual variation in food availability (*Poole & Graf, 1996*), and presumably requires martens to defend large amounts of space within their home ranges to meet

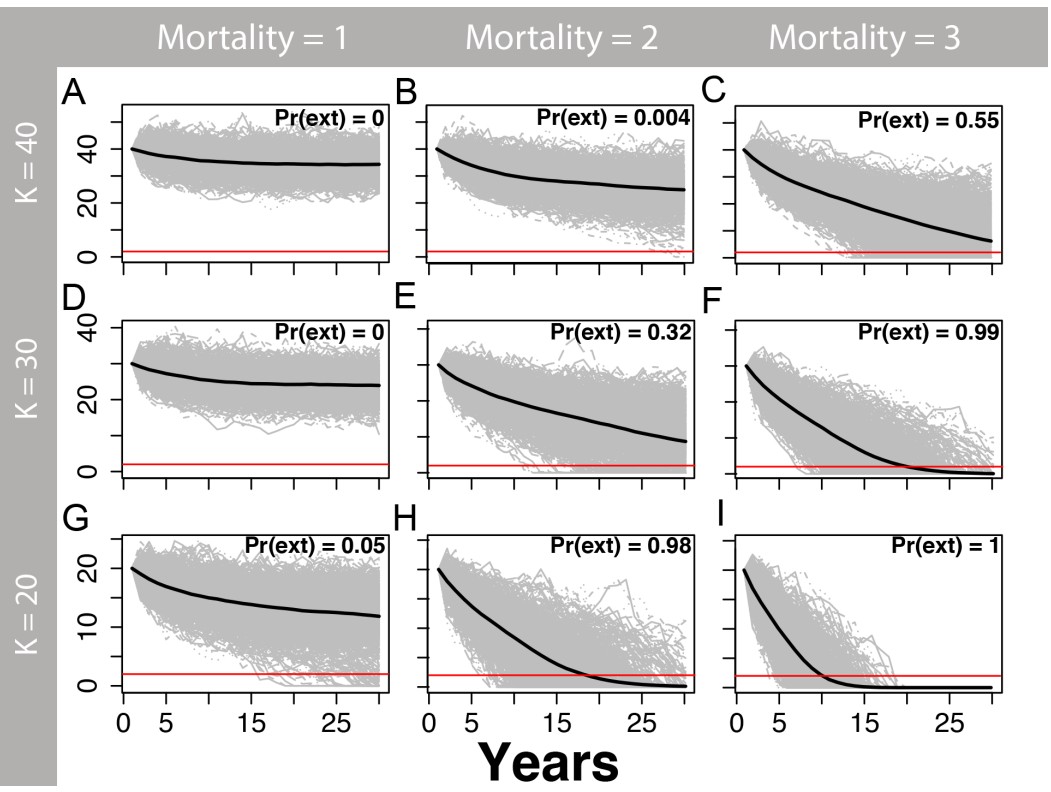

**Figure 5 Marten population viability analysis, theta = 1.** One-thousand density-dependent stochastic population projections (gray) for coastal Pacific marten (*Martes caurina humboldtensis*) from the theta-logistic model, assuming a linear relationship between per-capita population growth and population size (theta = 1) beginning at three values of carrying capacity ($K$; 20, 30, or 40), and three human-caused mortalities averaging 1, 2, or three marten annually. The mean population trajectory is given by the black line, and the red line signifies the pseudo-extinction threshold of 2 individuals. The proportion of trajectories falling below this threshold is the probability of extirpation Pr(ext). Stochastic mortalities averaging two or more marten lead to substantial extirpation risk within the next 30 years, particularly for smaller values of $K$.

nutritional requirements. In contrast, low-latitude coastal populations have a broad diet including foods such as late-season berries and over-wintering passerine birds, unavailable to montane and high latitude martens, particularly during winter (*Nagorsen, Morrison & Forsberg, 1989*) potentially facilitating the small home ranges observed in our study.

Despite the adjacent high-density marten population, the mature forest east of the Oregon Dunes does not support a marten population. The reason for near complete marten absence to the east is unclear. We hypothesize that abundant berry-producing shrubs directly provide abundant food for martens, and indirectly support marten by increasing the abundance of frugivorous vertebrate prey. Moreover, dense understory vegetation likely mediates interactions with competitors and predators, and provided spaces to hunt and avoid predators similar to snow in winter (*Andruskiw et al., 2008*). As such, maintaining contiguous tree and shrub cover—and limiting fragmentation and habitat loss—would benefit martens in areas where they persist. Whether prey availability,

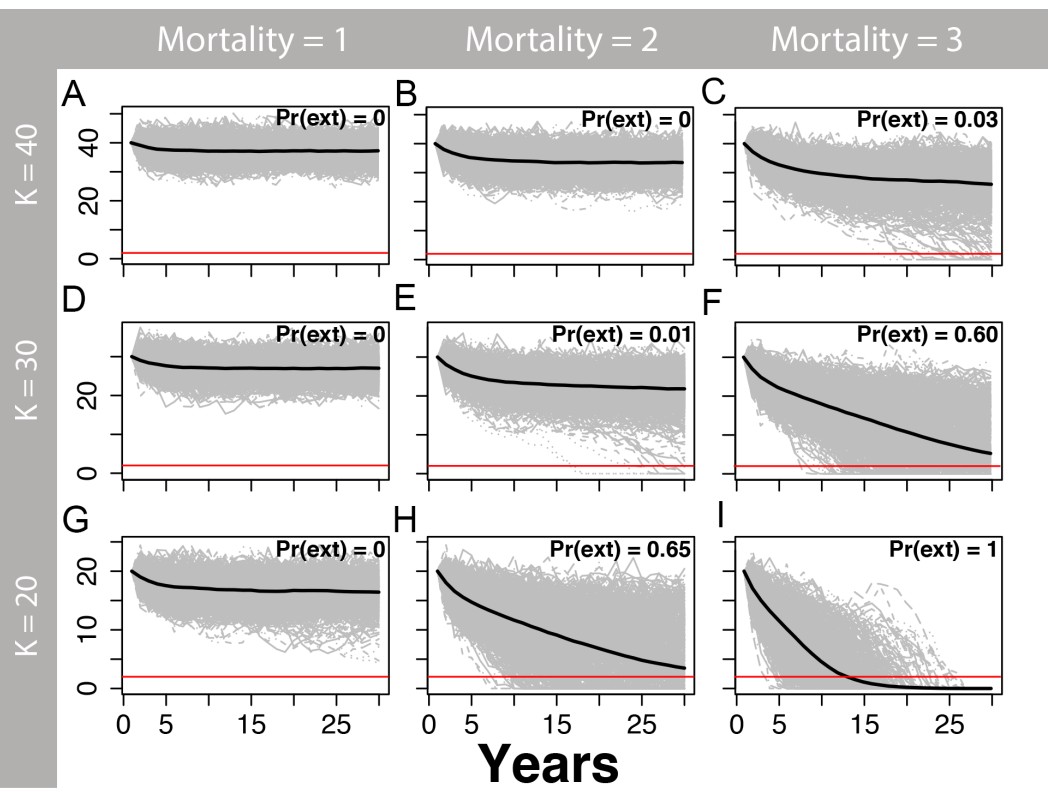

**Figure 6 Marten population viability analysis, theta = 2.** One-thousand density-dependent stochastic population projections (gray) for a coastal Pacific marten (*Martes caurina humboldtensis*) from the theta-logistic model assuming a convex relationship between per-capita population growth and population (theta = 2) beginning at three values of carrying capacity ($K$; 20, 30, or 40), and three human-caused mortalities averaging 1, 2, or three marten annually. A theta >1 may be more realistic for long-lived mammals, because the onset of density dependence likely occurs at higher population densities once crowding of territories occurs; assuming values where theta >1 is less conservative because the population will be more permissive to mortality or other mortality. The mean population trajectory is given by the black line, and the red line signifies the pseudo-extinction threshold of two individuals. The proportion of trajectories falling below this threshold is the probability of extirpation Pr(ext). Within the next 30 years, stochastic mortalities averaging two or more marten lead to substantial extirpation risk, particularly for smaller values of $K$.

habitat-mediated competition, or some combination of these factors limits martens from the extensive inland forests is largely unknown; these questions are key to address when considering the potential for population expansion.

In addition to vegetation structure and predation, harvest by humans can affect marten populations. It is currently legal to harvest marten throughout Oregon, including within this small, remnant, coastal population. Marten populations can be resilient to fur harvest when they are abundant, and if breeding females are harvested infrequently compared to males, particularly juvenile males (*Robitaille, 2017*; *Banci & Proulx, 1999*). Adult females in our study were observed more frequently than adult males at ratios of 1.5:1 (live-trapping) and 3:1 (SMR). Our results were atypical of ratios observed in other marten research studies and in harvested populations; these studies typically demonstrate higher male to female

ratios of live-trapped and kill-trapped martens (e.g., *McCann, Zollner & Gilbert, 2010*; *Payer & Harrison, 1999*; *Robitaille, 2017*). Given the small population size and vulnerability to trapping, eliminating fur harvest in the central coast of Oregon would decrease immediate risk of marten extirpation.

We have provided a baseline estimate of population size that can be compared to future surveys, allowing the monitoring of population status and viability. Such additional monitoring efforts would inform whether these populations are declining or merely small (*Caughley, 1994*). Small population size, consistent annual human-caused mortality, and isolation indicate this coastal marten population is likely to remain vulnerable to extirpation.

## ACKNOWLEDGEMENTS

We received considerable aid with field logistics, vehicles, housing, and equipment from the Central Coast Ranger District, Siuslaw National Forest. Adam Kotaich contributed significantly to the field work, we also thank Cindy Burns, Crystal Mullins, and Deanna Williams for quickly using field data for management-related discussions and the team that has incorporated martens into the updated Oregon Dunes Restoration Strategy.

### Funding
Survey efforts were funded by the USDA Forest Service Pacific Northwest Research Station, Siuslaw National Forest, and US Fish and Wildlife Service's Portland Office. The funders had no role in study design, data collection and analysis, decision to publish, or preparation of the manuscript.

### Grant Disclosures
The following grant information was disclosed by the authors:
USDA Forest Service Pacific Northwest Research Station.
Siuslaw National Forest.
US Fish and Wildlife Service's Portland Office.

### Competing Interests
The authors declare there are no competing interests.

### Author Contributions
- Mark A. Linnell conceived and designed the experiments, performed the experiments, analyzed the data, contributed reagents/materials/analysis tools, prepared figures and/or tables, authored or reviewed drafts of the paper, approved the final draft.
- Katie Moriarty conceived and designed the experiments, performed the experiments, contributed reagents/materials/analysis tools, prepared figures and/or tables, authored or reviewed drafts of the paper, approved the final draft.

- David S. Green and Taal Levi analyzed the data, contributed reagents/materials/analysis tools, prepared figures and/or tables, authored or reviewed drafts of the paper, approved the final draft.

## Animal Ethics

The following information was supplied relating to ethical approvals (i.e., approving body and any reference numbers):

The USDA Forest Service Institutional Animal Use and Care Committee provided approval for this research (Permit USFS 2015-002).

## Field Study Permissions

The following information was supplied relating to field study approvals (i.e., approving body and any reference numbers):

We conducted research with live animals approved by Oregon Department of Fish and Wildlife Department Scientific Take Permit (ODFW 119-15).

## Data Availability

The raw marten location data is not provided to reduce negative population impacts.

## Supplemental Information

Supplemental information for this article can be found online at http://dx.doi.org/10.7717/peerj.4530#supplemental-information.

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
