# Peer review of "Density and population viability of coastal marten: a rare and geographically isolated small carnivore"

_PeerJ, doi:10.7717/peerj.4530_

## Round 0.1 · original submission · Major Revisions

Dear Dr. Moriarty,

I have assessed your paper and the comments from two experts and believe that your manuscript requires major revisions before being acceptable for publication in PeerJ.

Reviewer 1 has highlighted extensive methodological considerations that you should rebut or address within your analyses. In particular, issues relating to unmarked observations and the use of telemetry in your spatial mark-resight models, the spatial configuration of your trapping effort and your habitat masking.

Reviewer 2 has highlighted some minor revisions for you to consider, in particular in relation to the autecology of the species.

Kind regards

Dr. Andrew Byrne

Reviewer 1 ·

Basic reporting

Generally looks fine to me (thanks, authors—written clearly is most places).

It might be helpful to include a layout of the trapping grid used for collaring, which I’ll discuss below.

L139—suggest “local convex hull (LoCoH)” and then using acronym rather than vice versa.

I suggest CRI rather than CrI as the acronym for Credible Intervals. In some places (e.g., L 249), you use plus/minus to denote a range—for clarity/ consistency, I would just stick with presenting CI or CRI for estimates rather than SE, and present means and range (rather than SD [? Not described]) when summarizing raw data.

L256: the t-stat and p-value suggest no difference, but I think also presenting estimates of territory size by region with CI would give the reader a little more information.

L290: The juxtaposition of the first two sentences in this paragraph is awkward. I think your point is that the observed anthropogenic mortalities, rather than evidence of a large population, instead suggest that the smaller population is at risk.

L301: Can you reword this? “Additional resolution” reads a little awkwardly.

L305: Rather than disproportionate, suggest saying greater or larger. Not sure why “nesting birds in plowed fields” needs to be explicitly included in this sentence.

Generally, the discussion seems a little less fully formed than other sections of the manuscript. I’ll revisit this in section three, but want to note it here because some issues with interpretation may simply be issues with the clarity and structuring of the writing.

From looking at the code and the text, it’s not clear to me how density is being estimated or when the masking is taking place: is D estimated as N-hat/Area with or without the masking of specific cells, or as the mean of the estimated point process intensity, or?

The data is set up in a standard flat format, which is fine and easier to store: for transparency, I might just add a comment to the code file indicating that the input data needs to be manipulated prior to model fitting.

Table 1: The input value for m here doesn’t seem to have any connection to the studies you cite—typo?

Table 4: Could you present p0 parameters as p0[male] and p0[female]? It’s not immediately clear which sex is which as presented.

Fig 1: Could you add an additional inset showing the distribution of cameras/traps? They don’t show up well here.

Fig 2: Instead of Mortality = x (which seems deterministic), how about Average Mortality = x?

Experimental design

I guess I interpret this part of the review as pertaining to methods/implementation. This is a clever design for answering the questions of interest, and the sampling effort (and the data collected) seem sufficient.

Several questions about how density was estimated. Throughout, I suggest describing this as a spatial mark-resight study rather than spatial mark-recapture (the same acronym is fine) because the model incorporates unmarked and marked detections.

1) A more detailed description of how unmarked observations and telemetry observations were integrated within the text is necessary for a reader to figure out what was performed without digging into code. Royle et al. use telemetry to get a better sense of habitat use (e.g., habitat/environment based changes in lam0 related to third order resource selection), while Sollmann et al. use telemetry exclusively to derive sigma (as performed here). The description of how of unmarked detections were integrated is similarly opaque (but seems to follow the likelihood described by Chandler and Royle in the 2013 AoAS paper). Please include some fundamental parts of the likelihood pertaining to these components of the model.

2) Could you provide more detail regarding the spatial configuration of the live-trapping effort in relation to the camera trapping effort? One assumption of SMR models is that the spatial distribution of marked and unmarked individuals is essentially the same, but it’s tough to evaluate this without a sense of where the live-traps were placed. My sense is that this assumption is being violated (it seems like marked detections far outweigh unmarked detections, and that marked animals are much more exposed to sampling than unmarked animals), and it can lead to bias/poor coverage (see recent paper in Journal of Applied Ecology by Jesse Whittington et al. regarding “Generalized Spatial Mark-Resight” for a description of the issues and models to deal with them).

3) I’m generally all for integration of disparate data sources to improve estimate precision. But one risk of integrating different data sources to estimate a shared parameter is that if the parameter is not ‘shared’ between datasets, you’re going to get a (precisely) biased estimate (There’s a paper “Balancing precision and risk” by Tabitha Graves et al. [2012] in Plos One that describes this nicely). I have some misgivings about integrating the telemetry and detection data in this case because I’d expect that baiting at cameras might be altering animal space use. Previous studies have suggested that incorporating telemetry data from a few individuals has basically no effect on estimates of density, but in this case, most of the data used to estimate sigma are derived from telemetry data: if ‘telemetry sigma’ is actually smaller than ‘trap sigma’ (which is what I'd surmise), shrinkage in estimates of the shared parameter might lead to overestimating marten density. Although Viorel Popescu’s 2014 paper with fishers implies that there probably is no consequence of combining these data types, I guess I would feel more comfortable with integrating the telemetry data if you could provide some evaluation of their consistency or sensitivity to their consistency. Maybe the easiest way to evaluate sensitivity is to see whether estimates change very much if the telemetry data are removed.

4) The observation likelihood is off. Y is treated and described as a Poisson RV, but you seem to be imposing restraints to make it behave like a Binomial RV wherever possible (e.g., you’re placing a natural limit on Y based upon the number of days a camera operated because defining Y as the number of days in which an individual was detected [the exact # successes in K trials that defines the Binomial distribution], and modeling variability in the intercept using a logit function). My guess is that you’re forcing this as a Poisson observation model to allow the unmarked data to be integrated following Chandler and Royle 2013—but that observation model isn’t necessary to integrate the unmarked data. I’d use a Binomial/Bernoulli observation model instead: the unmarked component of the model should look a little more like the what is described by Chandler and Clark 2014 (Methods in Ecology and Evolution 5[12]) or David Ramsey et al. 2015 in the Journal of Wildlife Management (79[3]). For consistency with the literature, use p[i,j,k] and p0 rather than lamda and lam0 when using a Binomial/Bernoulli observation model.

5) Because the stations are baited, I think that incorporating a behavioral response to initial capture (probably a trap-specific behavioral response) using an indicator covariate is probably important. This is typically modeled using a Bernoulli encounter model. The Bernoulli likelihood will take a little longer to run, particularly with an imhomogenous point process model—here's a description of a tip that might help speed things up (https://groups.google.com/forum/#!topic/spatialcapturerecapture/v4KA8rSuX7Y). One caveat: in my limited experience working with unmarked Spatial Count type models, estimating a behavioral response has been tricky, and it may be impossible without additional information. I think the marked animals should provide sufficient info to allow a behavioral response to be modeled, but if you can’t, I think presenting your results as conservative estimates of population size is probably ok.

6) The chains used to generate MCMC samples seem short—in particular, the burn-in of 500 iterations seems insufficient. Are you sure that you are sampling from the stationary posterior distribution? The Geweke diagnostic might be useful for checking this.

7) Please provide a little more clarity regarding masking out habitat. As far as I can tell, you fit the inhomogenous point-process model without masking out anything, but masked out non-habitat when predicting? I think it is reasonable to mask out sand-dunes when predicting density into the other region (it doesn’t look like the camera transect is really set up to capture the effect of forest cover on marten density all that effectively), but as noted above, it’s a little difficult to figure out just how abundance and density estimates presented are is being derived here (it seems like it’s derived from mu[g]). As noted below, if estimates reflect masking habitat out after the fact, comparison with other studies is a little trickier.

Territory mapping: My sense is that demonstrating what you did within a graphic (perhaps adding onto to Fig. 3) might make interpretation easier. There seems to be disagreement within the text over what the territory mapping is estimating (carrying capacity, or actual population size). I’m not really convinced that this is a great method for doing either, and I don't see this as a necessary part of the paper, but maybe there's a compelling reason for keeping it that I'm missing. Regardless of how you prefer to interpret these predictions, if you keep this part, I think some discussion of the assumptions being made about marten space use and some justification for these assumptions is probably warranted.

PVA: generally seems fine, although I’m not qualified to review this with any technical depth. A couple minor questions regarding clarity/implementation. Why treat human-caused deaths as a Poisson RV, if only to later argue that anthropogenic mortality is not stochastic (Line 300)?

Given that the Poisson is skewed, maybe extinction at higher expected values is really being driven by sporadic but extreme mortality events? The modeled variation in anthropogenic mortality exceeds the range of observed harvest…is there sufficient empirical data to inform an alternative generating distribution for this variable? Could you add a little justification for choosing lambda values of 1,2,3 (e.g., we select these values to explore sensitivity to limited increases, or we assume that baseline mortality is higher because roadkill may go unreported, etc.)?

Could you provide a more detailed justification for assuming these population as lacking immigration in the methods (on L312 you note barriers, distance, etc., but can these things be quantified and presented in the methods)?

Finally, I think might be useful to incorporate uncertainty into the PVA projections (I think confidence intervals could be derived from non-parametric bootstrapping).

Validity of the findings

I guess this section pertains to presentation/interpretation of the results.

L269: As presented in the methods, the territory mapping was used to estimate marten carrying capacity rather than abundance; here it is treated as an additional technique for estimating population size.

L278 & L285: I’d say 2-3 annual mortalities on average instead of 2-3 mortalities.

L299-315: Maybe this point belongs under basic reporting, but the focus here is unclear. The take-home of the PVA is that relatively low levels of anthropogenic mortality put these populations at risk. I’m not sure why juvenile recruitment is being mentioned here as another potential factor, and then summarily dismissed. I'd suggest focusing upon your own results a little more closely throughout the discussion.

L316-329: Two comments. First, I’m not sure that density estimates based on masking out non-suitable habitat are really comparable to density estimates reported by other studies. You’re reporting density per sq. km of suitable habitat; other studies typically report density per sq. km of land. I’d at least note that this is likely to amplify your estimates relative to other systems.

Secondly, do you think you could expand upon upon some of your predictions and alternatives here? An alternative is that rather than the distinct vegetation in this system mediating marten co-existence with other animal species, the animal community in the system is distinct (as you mention, year-round passerine birds). Or perhaps sympatric predators are less abundant in this system—martens in Newfoundland readily use beetle-kill and clearcuts that provide little cover, but sympatric predators are relatively rare (Hearn et al. 2010, Journal of Wildlife Management 74[4]). Presumably, the cameras provide some information about other carnivores in the study area?

L330-L335: I suggest avoiding any discussion of trying to manage inland forests to resemble coastal forests.

335-338: This is really the main finding of the study: this is an important but vulnerable marten population. I suggest that rather than ending with this sentence, you shift it earlier into the discussion and talk a little bit about how maintaining habitat and limiting mortality might be accomplished.

·

Basic reporting

This is a well-written article. the authors used up-to-date references. I suggest that Buskirk et al (2012) in Aubry et al.'s book, be cited to support statements in lines 314-315.

Experimental design

This is a well-designed scientific research with a proper inductive review of the problem. There is a bit of confusion regarding the habitat preferences of the Humboldt marten. The introduction indicates that the species is found in late-seral stages - and such stages are present in federal parks where the habitats are protected. However, there were no marten surveys in these parks. The martens are apparently found in 70-yr-old forests (study area). So it is incorrect to conclude that the species is a late-seral specialist (line 64). So the authors need to explain the discrepancy - either the species is wrongly believed to be a late-seral species, or the martens may use younger forests (as in the study area) - in the latter case, th presence of martens in young forests may be due to the fact that there is no other habitat at regional level or the food items are very abundant and the species is more flexible in its choice of habitats.
Methods are properly detailed. However, lease indicate the models and manufacturers of remote cameras.
Why do they authors refere to "territory" instead of "home range"?

Validity of the findings

The conclusions follow the data. The findings are significant and may play an important role in the conservation of the species

Additional comments

Good work. the marten-habitat relationship needs to be better described and explained in the introduction. Please consider the following:

Line 58, martens instead of marten.
Line 93 - The Oregon Dunes consist of...
Line 131 - 5 min instead of 5 minutes
Line 133 - hrs instead of hours.
Line 184 - within one (or 1) hour - no hyphen.
Lines 252 and 275 - >40% forest cover - please specify the cover (age and structure).
General note - your argument that, 'because of human-caused mortalities and limited habitat, the future of marten population may be jeopardized' relates to the level of resiliency of the species, a concept that has been developed by Banci and Proulx (1999- Mammal Trapping published by Alpha Wildlife Publications).

---

## Round 0.2 · Minor Revisions

Your paper has been reviewed again by both reviewers, and both recommend a minor revision to the manuscript. However, given the extensive nature of the technical comments made by reviewer 1 (Dr. John Clare), the manuscript may require additional analysis/rerunning of models. Please either clarify, address or rebut the points raised by Dr. Clare, or indicate whether further exploration of your dataset will be undertaken in other MSs.

Reviewer 2 highlights an issue around trapping of the martens, which should be commented on, or rebutted.

Both reviewers, and myself, commend the extensive work done on the paper after the first round of reviews.

Reviewer 1 ·

Basic reporting

The one minor issue is that it looks like sigma and maybe some other greek symbols are being altered, which makes following some of the formulas difficult. For example, I can't quite figure out what's happening on line 187.

Experimental design

Thanks for the work put in here to alter some of my previous concerns. I just want to frame the following by noting that these are pretty advanced models that can be difficult to implement, I appreciate the effort the authors are putting into this, and I don't want to hang things up for no reason. With that said, two comments/questions:

1) Regarding the marking process (y.trap): you're sure that baseline encounter probability is constant? There's no trap-happiness or trap-shyness associated with actual trapping (as was implemented for the camera resighting)?

2) More substantively, the resighting process. A) In looking at the code and what I can make of the formulas in the text, you have a resighting process that consists of i) marked individuals that can always be identified, and ii) unmarked individuals that cannot be distinguished, but not iii) marked individuals that cannot be distinguished, correct? (Whittington et al. dealt with all three, but if there is no iii, the general structure here looks fine).

B) This is going to a big blob of text for which I apologize; I don't intend to be pedantic, but want to make sure you are comfortable with how you are modeling this and make sure that the model is valid, so:

There are basically two ways to model the unmarked component. In the first, your unmarked data are basically counts of detections during specific intervals, where each latent encounter history is Poisson distributed (an individual can be detected multiple times). The station-specific detection total per interval or overall can be treated as the sum of these latent individual Poisson variables (or you could marginalize over the latent encounter histories). Something like:


lambdaresight[i,j,k] <- lam0.resight[i,j,k]*exp(-d.resight2[i,j]/(2*sigma2[Sex2]))*z*(1 - marked[i]))
log(lam0.resight[i,j,k]) <- delta0 + delta1*Sex[i] + delta2*Baiting[j,k]
countu[i,j,k] ~ dpois(lambdaresight[i,j,k]
nU[j,k]~dsum(countu[1, j, k], countu[2, j, k].....etc.)


Alternatively, you might just look at presence-absence by station (i.e., there's >0 detections or not), in which case the likelihood looks a little different: the underlying encounter model for an individual follows the bernoulli model you have here, with the recorded presence-absences at a station = 1-(1-punmarked[i, j, k])^Nunmarked. Something like:

p_unmarked[i,j,k] <- lam0.resight[i,j,k]*exp(-d.resight2[i,j]/(2*sigma2[Sex2]))*z*(1 - marked[i]))
logit(lam0.resight[i,j,k]) <- delta0 + delta1*Sex[i] + delta2*Baiting[j,k]
Ptrap[j, k]<-1-prod(1-p_unmarked[,j,k])
yu[j, k]<-dbern(Ptrap[j, k])


As far as I can tell, you are treating the total # of unmarked marten pictures as arising from the sum of latent Bernoulli encounter histories. This is potentially problematic, because it implies that x marten detections = x martens were detected. A little more detail regarding how the unmarked data are defined (an image? an image sequence? >0 images during a day?) would be useful, and should dictate which formulation you use.

The data file suggests there is never >1 unmarked detection during an interval. Depending upon how the unmarked data are defined, this might mean that your formulation works okay, although you might wish to note that your formulation is not going to be correct in many circumstances.

I don't want to eat up a ton of space getting in the weeds here: I'll sign this, and if anything does not make sense or could use clarity, please reach out and let me know.

John Clare
University of Wisconsin
jclare2*at*wisc.edu

Validity of the findings

No issues.

Additional comments

The authors should be commended for putting a lot of work into revising the submission, and more generally, for attempting to tackle the underlying study problem with a high level of technical rigor. This is nice work, and I appreciate the thorough response to previous comments.

·

Basic reporting

Clear and unambiguous.

Experimental design

The text benefited from the comments expressed by Referee # 1 in the first review.

Validity of the findings

Very important findings.

Additional comments

I have a few comments only.
1. Considering that trapping may impact significantly on the population survival, I recommend that all forms of trapping be eliminated within and outside the area of concern. Accommodating trappers on the outside (so you capture more male juveniles than females) can still impact on the marten population in your study area. Martens have the ability to find baits set far away and these baits on the outside could cause displacement of some animals. Trap sets outside the study area could still impact on the population survival. Since we are talking about survival of a species, measures should be taken to avoid all deaths. You may not be able to stop vehicles, but you can remove trapping.

2. The reference "Banci and Proulx 1999" is missing in Literature Cited:
Banci, V., and G. Proulx. 1999. Resiliency of furbearers to trapping in Canada. Pages 175-203 in G. Proulx, editor, Mammal trapping, Alpha Wildlife Publications, Sherwood Park, Alberta.

3. Figure 1, legend = The verb tenses need to be reviewed. For example, you used the past tense in "the study area was bonded..." and the present tense for "This area has extensive fragmentation...". I suggest the authors harmonize the verbs, and use the proper tenses. For example, the study area is bonded (and it still is!), but the area had extensive fragmentation (at time of study).

4. Figure 2, legend. I think that lines 535-541 should be included in Methods, i.e., line 152.

5. Line 544 - Home ranges or territories?

---

## Round 0.3 · accepted · Accept

Having reviewed your responses to reviewers comments, I am content that you have addressed and/or explored potential issues raised.